# High Betaine, a Trimethylamine N-Oxide Related Metabolite, Is Prospectively Associated with Low Future Risk of Type 2 Diabetes Mellitus in the PREVEND Study

**DOI:** 10.3390/jcm8111813

**Published:** 2019-11-01

**Authors:** Erwin Garcia, Maryse C. J. Osté, Dennis W. Bennett, Elias J. Jeyarajah, Irina Shalaurova, Eke G. Gruppen, Stanley L. Hazen, James D. Otvos, Stephan J. L. Bakker, Robin P.F. Dullaart, Margery A. Connelly

**Affiliations:** 1Laboratory Corporation of America Holdings (LabCorp), Morrisville, NC 27560, USA; dwbent@uwm.edu (D.W.B.); eliasjey@gmail.com (E.J.J.); shalaui@labcorp.com (I.S.); otvosj@labcorp.com (J.D.O.); connem5@labcorp.com (M.A.C.); 2Department of Nephrology, University of Groningen and University Medical Center Groningen, 9700 RB Groningen, The Netherlands; m.c.j.oste@umcg.nl (M.C.J.O.); e.g.gruppen@umcg.nl (E.G.G.); s.j.l.bakker@umcg.nl (S.J.L.B.); 3Department of Endocrinology, University of Groningen and University Medical Center Groningen, 9700 RB Groningen, The Netherlands; r.p.f.dullaart@umcg.nl; 4Department of Cardiovascular and Metabolic Sciences, Cleveland Clinic, Cleveland, OH 44195, USA; HAZENS@ccf.org; 5Department of Cardiovascular Medicine, Cleveland Clinic, Cleveland, OH 44195, USA

**Keywords:** betaine, trimethylamine N-oxide related metabolites, nuclear magnetic resonance spectroscopy, type 2 diabetes mellitus

## Abstract

Background: Gut microbiota-related metabolites, trimethylamine-N-oxide (TMAO), choline, and betaine, have been shown to be associated with cardiovascular disease (CVD) risk. Moreover, lower plasma betaine concentrations have been reported in subjects with type 2 diabetes mellitus (T2DM). However, few studies have explored the association of betaine with incident T2DM, especially in the general population. The goals of this study were to evaluate the performance of a newly developed betaine assay and to prospectively explore the potential clinical associations of betaine and future risk of T2DM in a large population-based cohort. Methods: We developed a high-throughput, nuclear magnetic resonance (NMR) spectroscopy procedure for acquiring spectra that allow for the accurate quantification of plasma/serum betaine and TMAO. Assay performance for betaine quantification was assessed and Cox proportional hazards regression was employed to evaluate the association of betaine with incident T2DM in 4336 participants in the Prevention of Renal and Vascular End-Stage Disease (PREVEND) study. Results: Betaine assay results were linear (y = 1.02X − 3.75) over a wide range of concentrations (26.0–1135 µM). The limit of blank (LOB), limit of detection (LOD) and limit of quantitation (LOQ) were 6.4, 8.9, and 13.2 µM, respectively. Coefficients of variation for intra- and inter-assay precision ranged from 1.5–4.3% and 2.5–5.5%, respectively. Deming regression analysis of results produced by NMR and liquid chromatography coupled to tandem mass spectrometry(LC-MS/MS) revealed an R^2^ value of 0.94 (Y = 1.08x – 1.89) and a small bias for higher values by NMR. The reference interval, in a cohort of apparently healthy adult participants (*n* = 501), was determined to be 23.8 to 74.7 µM (mean of 42.9 ± 12.6 µM). In the PREVEND study (*n* = 4336, excluding subjects with T2DM at baseline), higher betaine was associated with older age and lower body mass index, total cholesterol, triglycerides, and hsCRP. During a median follow-up of 7.3 (interquartile range (IQR), 5.9–7.7) years, 224 new T2DM cases were ascertained. Cox proportional hazards regression models revealed that the highest tertile of betaine was associated with a lower incidence of T2DM. Hazard ratio (HR) for the crude model was 0.61 (95% CI: 0.44–0.85, *p* = 0.004). The association remained significant even after adjusting for multiple clinical covariates and T2DM risk factors, including fasting glucose. HR for the fully-adjusted model was 0.50 (95% CI: 0.32–0.80, *p* = 0.003). Conclusions: The newly developed NMR-based betaine assay exhibits performance characteristics that are consistent with usage in the clinical laboratory. Betaine levels may be useful for assessing the risk of future T2DM.

## 1. Introduction

Betaine (N,N,N-trimethylglycine) is an osmoprotectant as well as a methyl donor in one-carbon metabolism [1,2]. Betaine is required for remethylation of homocysteine to methionine, a precursor of the universal methyl donor S-adenosylmethionine (SAM) [1,2]. By decreasing SAM availability, betaine deficiency may decrease phosphatidylcholine synthesis, promote hepatic steatosis and modify very low-density lipoprotein (VLDL) synthesis and secretion [3]. Betaine was shown to be inversely associated with triglycerides (TG) and phospholipid transfer protein (PLTP) activity, further supporting the notion that low betaine levels may alter liver fat accumulation and lipid/lipoprotein metabolism [4]. Lower plasma betaine concentrations have been reported in subjects with metabolic syndrome, type 2 diabetes mellitus (T2DM), non-alcoholic fatty liver disease (NAFLD) and/or non-alcoholic steatohepatitis (NASH) [3,5,6,7,8]. Moreover, betaine levels were associated with disease severity in subjects with NAFLD, betaine levels being significantly lower in subjects with NASH compared to subjects with hepatic steatosis [3]. Plasma betaine levels arise from both dietary intake and endogenous synthesis from dietary choline [1,2,9]. Having a high-throughput assay to measure plasma/serum betaine levels may be a way of gaining a better understanding of the role of betaine in hepatic and metabolic diseases.

The gut microbe-derived metabolite TMAO has been shown to be an independent marker of cardiovascular disease (CVD) and mortality with higher levels of TMAO correlating with higher risk [10,11,12,13,14,15,16,17,18,19,20,21,22]. Betaine has also been reported to be associated with CVD. However, in contrast to TMAO, low plasma betaine levels were associated with increased CVD risk in most studies [14,23,24,25,26]. In one study, however, Lever et al. reported that low betaine levels were associated with CVD events in patients without T2DM, whereas in contrast, elevated betaine levels were associated with CVD in patients with T2DM [27]. In addition, it has been shown that the association of betaine with CVD risk was attenuated by TMAO, suggesting that it is TMAO that drives the risk of future CVD [14]. In the same cohort [14], it was also found that a modest but statistically significant correlation exists between TMAO and betaine.

Besides the cross-sectional associations of betaine with metabolic diseases, as well as prospective relationships of betaine with CVD events, prospective studies have shown that lower betaine levels were associated with a higher risk of incident T2DM. In a study of Norwegian subjects with suspected angina pectoris, plasma betaine levels were inversely associated with incident T2DM even after adjusting for multiple clinical characteristics and risk factors [7]. In subjects enrolled in the Prevención con Dieta Mediterránea (PREDIMED) trial, higher betaine levels were associated with a reduced risk of future T2DM [28]. In the Diabetes Prevention Program (DPP), a program designed to study progression to T2DM in subjects with elevated fasting plasma glucose (FPG) and impaired glucose tolerance, low betaine levels were associated with T2DM onset [6]. Moreover, this association remained significant even after adjusting for covariates such as age, sex, body mass index (BMI), hypertension, ethnicity, and FPG [6]. Additionally, the DPP lifestyle intervention led to an increase in betaine levels, suggesting that betaine may have utility in monitoring the effects of interventions designed to prevent progression to T2DM [6]. However, no studies to date have explored the association of betaine with incident T2DM in a general population-based cohort. 

The goals of this study were to evaluate the laboratory performance of the newly developed nuclear magnetic resonance spectroscopy (NMR)-based betaine assay and to explore potential clinical associations of betaine and incident T2DM in a large prospective population-based cohort.

## 2. Methods

### 2.1. NMR Data Acquisition and Betaine Quantification by Peak Deconvolution

Serum/plasma specimens were diluted with citrate/phosphate buffer (3:1 v/v) to lower the pH to 5.3 in order to separate the betaine and TMAO signals which overlap at physiological pH (Figure 1). One-dimensional (1D) 1H-NMR spectra were recorded as previously described [4,29]. Briefly, the Carr–Purcell–Meiboom–Gill (CPMG) acquisition technique was utilized to suppress resonances from macromolecules (e.g., proteins and lipoproteins) along with Water suppression Enhanced through T_1_ effects (WET) gradient sequence to attenuate the water signal. Spectra were acquired at 47 °C (spectral width = 4496.4 Hz, relaxation delay between scans = 5 s, direct detection time = 1.2 s, number of scans = 48) on a Vantera NMR clinical analyzer (LabCorp, Morrisville, NC) equipped with 400 MHz (9.4 T) Agilent spectrometer [29,30]. Betaine was quantified from the acquired spectra using a proprietary deconvolution algorithm that resolved the betaine region into its spectral components (Figure 1). The α-anomeric glucose signal is used as a reference to locate the betaine peak. The betaine peak at 3.22 ppm, as well as the glucose peaks on either side, were mathematically modeled using lognormal and Lorentzian peak shapes. A linear function was incorporated into the algorithm to adequately model the baseline around these peaks. After subtracting the baseline, the lineshape deconvolution was achieved by a bound-constrained non-linear least-squares fitting algorithm. The derived betaine signal amplitudes were converted to µM units using a factor that was empirically determined by plotting data from dialyzed serum samples spiked with known amounts of betaine and relating the betaine signal area to the expected concentrations.

### 2.2. Assay Performance Testing

All assay performance studies were conducted according to Clinical and Laboratory Standards Institute (CLSI) guidelines. Dialyzed serum pools devoid of betaine were analyzed as blanks (five pools, four replicates, three days). Serum pools containing low concentrations of betaine were tested to determine the limits of detection (LOD) (five pools, four replicates, three days) and quantitation (LOQ) (eight pools, four replicates, three days). The LOB, LOD, and LOQ were calculated as previously described [30]. Within-run (*n* = 20 replicates) and within-laboratory (*n* = 80 replicates) imprecision were determined using serum pools with low, intermediate and high betaine values. These values were targeted around the 50th and 100th percentile of the reference interval, as well a high value that had been observed in clinical samples. Mean concentrations and coefficients of variation (%CV) were calculated for each pool. For the tube comparison, specimens were drawn from 21 donors into three blood collection tube types: Serum LipoTubes (Greiner Bio-One, Monroe, NC, USA), K_2_EDTA plasma tubes and plain red top serum tubes without gel barriers (BD Diagnostics, Durham, NC, USA). Specimen tubes were processed as per the manufacturer’s recommendations. Results for EDTA plasma and plain serum tubes were compared to results for the LipoTube by calculating the mean % bias for all 21 samples. Linearity was evaluated by comparing known spiked concentrations of betaine with expected concentrations tested in quadruplicates across the biological range from 25 to 1200 μM (*n* = 13).

A method comparison study was performed to compare betaine quantification by NMR versus mass spectrometry as per CLSI guidelines [31]. Serum specimens were obtained from multiple donors and aliquots were frozen at -80°C until the time of analysis. The same frozen serum samples (*n* = 24) were analyzed via NMR at LabCorp and in parallel by stable isotope dilution, high-performance liquid chromatography with online electrospray ionization tandem mass spectrometry (LC-MS/MS) at the Cleveland Clinic. Samples analyzed via the LC-MS/MS method were injected at a flow rate of 0.8 ml/min to a 4.6 × 250 mm, 5 µm Luna silica column interfaced with an AB SCIEX 5000 triple quadrupole mass spectrometer using d9-(trimethyl)-labeled internal standards [32]. Details of the method for the LC-MS/MS assay have been previously described [10,32,33]. Samples for the method comparison study spanned the reported reference interval for betaine [1]. 

To confirm the previously reported reference interval for betaine [1], two studies were evaluated. For the first study, normal, apparently healthy (*n* = 501) men and women aged 18 to 80 were recruited at LipoScience (now LabCorp, Morrisville, NC, USA). Informed consent was obtained from all donors whose samples were analyzed and the study protocol was approved by a local Institutional Review Board. Non-fasting serum specimens collected in LipoTubes and tested by NMR as described above. Detailed descriptions of this study have been previously described [29,30]. The second study entailed reanalyzing the digitally stored NMR spectra, from previously tested fasting EDTA plasma samples from the Prevention of Renal and Vascular End-Stage Disease (PREVEND) Study (*n* = 5621), using the newly developed proprietary betaine assay software (LabCorp, Morrisville, NC, USA) [29]. Quantiles for sample results from both studies were determined, and the reference intervals were estimated at the 2.5th and 97.5th percentiles. Pearson’s correlation coefficients were determined, and no statistically significant correlation was found between betaine and TMAO in either the normal healthy population study that was used for the reference interval determination (*r* = 0.065, *p* = 0.149) or the PREVEND study (*r* = -0.01, *p* = 0.508).

### 2.3. Cross-sectional and Prospective Analyses in Participants in the Prevention of Renal and Vascular End-Stage Disease (PREVEND) Study

Details of the PREVEND study design and recruitment have been described before [34]. Briefly, the PREVEND study is a Dutch cohort of predominantly Caucasian men and women drawn from the general population of the city of Groningen in the northern part of the Netherlands. After exclusion of subjects with insulin-treated diabetes and pregnant women, subjects with a urinary albumin concentration ≥10 mg/L were invited to participate (*n* = 7768), 6000 accepted. In addition, a random sample of 2592 individuals with a urinary albumin concentration <10 mg/L was included. During the years 1997–1998, 8592 subjects aged 28–75 years completed the baseline survey. The baseline for the current study was the second screening which took place between 2001 and 2003 (*n* = 6892). Individuals who were diagnosed with T2DM or were missing data for diabetes at baseline or follow-up, as well as those missing NMR at baseline or follow-up, were excluded, leaving 4336 subjects for the present cross-sectional and prospective analyses. The follow-up period (period between baseline and date of T2DM diagnosis) was determined to be 7.3 (IQR, 5.9–7.7) years for this analysis. T2DM was ascertained if one or more of the following criteria were met: 1) FPG ≥7.0 mmol/L (126 mg/dL), 2) random sample plasma glucose ≥11.1 mmol/L (200 mg/dL), 3) self-report of a physician diagnosis of T2DM, and 4) initiation of glucose-lowering medication use, retrieved from a central pharmacy registry [35]. These criteria are consistent with current guidelines for the diagnosis of T2DM [36]. Calculations of BMI, as well as determinations of blood pressure, smoking status, alcohol intake, hypertension and estimated glomerular filtration rate (eGFR), have been described previously [37]. At the second screening, venous blood was obtained after an overnight fast. EDTA plasma samples were prepared and stored at −80 °C until testing occurred. Total cholesterol was measured on a Beckman Coulter® AU680 analyzer, and high-density lipoprotein cholesterol (HDL-C) and triglycerides (TG) were measured on an Olympus AU400 analyzer (Beckman Coulter, Brea, CA, USA) [38]. FPG was measured by dry chemistry (Eastman Kodak, Rochester, NY, USA) and C-reactive protein (CRP) was measured by nephelometry with a threshold of 0.18 mg/L (BNII, Dade Behring). Betaine results were obtained as described above. This study was carried out in accordance with The Code of Ethics of the World Medical Association (Declaration of Helsinki), cleared by an Institutional Review Board and all donors signed consent forms. 

### 2.4. Statistical Analyses

Statistical analyses were performed using SAS v9.4 (SAS Institute, Cary, NC, USA), Analyze-it v3.90.1 (Analyze-it Software, Ltd., Leeds, UK) or IBM Statistics SPSS v23.0 (IBM Inc., Chicago, IL, USA). For the analytical validation studies, linear regression analyses were performed. Deming regression analysis and Bland–Altman plots were used to evaluate the correlation between the results obtained on the two platforms for the method comparison study. Reference intervals were compared by assessing the means using the Wilcoxon Rank Sum and Mann–Whitney tests and the distributions using the Kruskal-Wallis test. For the epidemiological studies, data are expressed in mean ± SD (or SEM for figures) when normally distributed, median (interquartile range) for skewed distribution and number and percentages in case of categorical data. Differences between the tertiles of betaine were tested by ANOVA or Kruskal Wallis for continuous variables and with χ^2^- test for categorical variables. Pearson’s correlation coefficients were used to evaluate any potential relation between betaine and TMAO.

Crude and multivariable Cox proportional hazards regression analyses were performed to estimate the effect of betaine on T2DM in a crude analysis as well as analyses adjusted for age and sex (model 1). Further adjustments were then made for eGFR (model 2), BMI and smoking status (model 3), ethnicity, FPG, total cholesterol, HDL-C, TG, CRP, and use of lipid-lowering drugs (model 4). In continuous Cox proportional hazards regression models, betaine was log-base 2 transformed to allow for the expression of the hazard ratios (HRs) per doubling of betaine. Additionally, betaine was used as a categorical variable for analyses by tertiles. Data were presented as HRs and 95% CI confidence intervals (CIs). Potential effect modification was evaluated by age, sex, and eGFR. Potential effect modifiers were tested by entering both main effects and the cross-product term in Cox regression analyses. When effect modification was observed, we proceeded with stratified analyses. For the sensitivity analyses, we performed crude and multivariable Cox proportional hazards regression analyses excluding subjects with previous CVD, use of lipid-lowering drugs, microalbuminuria and eGFR <60 mL/min/1.73 m^2^ at baseline. Two-sided *p*-values <0.05 were considered statistically significant for all reported analyses.

### 2.5. Ethic Approval and Consent to Participate 

This study was carried out in accordance with The Code of Ethics of the World Medical Association (Declaration of Helsinki), cleared by an Institutional Review Board and all donors signed consent forms.

## 3. Results

The NMR signals from betaine and TMAO overlap in the Carr-Purcell-Meiboom-Gill (CPMG) collected spectrum of serum at physiological pH [29]. Hence, an assay was developed that reduces the pH of the sample to 5.3. The betaine peak is not affected by the change in pH, however, the TMAO peak is shifted downfield in the NMR spectrum which allows for accurate quantification of both betaine and TMAO separately [29] (Figure 1). In order to assess its clinical usefulness, the analytical performance of the NMR-based betaine assay was evaluated. The LOB, LOD, and LOQ were determined to be 6.4, 8.9, and 13.2 µM, respectively, with the LOQ being below the previously reported normal reference interval for betaine [1]. The coefficients of variation for intra-assay and inter-assay precision were 1.5–4.3% and 2.5–5.5%, respectively (Table 1). A tube comparison study revealed no significant differences between results from serum collected in LipoTubes vs. results from serum collected in plain red top tubes (mean % bias = −1.9) or EDTA plasma (mean % bias = 4.1). Linearity was demonstrated between 26.0 and 1135 µM, well above the upper limit of the normal reference interval [1], with a correlation coefficient (R^2^) of 1.00 and a linear equation of Y = 1.02x − 3.75 (Figure 2A). Based on these data, the reportable range for the NMR-based betaine assay was determined to be 13.2–1135 µM. 

A method comparison study was performed to compare the quantification of betaine by NMR versus LC-MS/MS. Deming regression analysis of betaine results from both platforms produced an R^2^ value of 0.94 (Y = 1.08x – 1.89) (Figure 2B). The Bland–Altman plot revealed that there was a small but systematic bias for higher values produced by the NMR assay compared to results from the LC-MS/MS assay (Figure 2C). The normal reference interval for betaine was confirmed in two populations, a cohort of apparently healthy volunteers and the full set of PREVEND cohort participants (Table 2). The distributions were similar between the two study populations, and the reference intervals were similar to those previously reported [1]. The distributions for men and women in the cohort of apparently healthy volunteers were not statistically different, and neither were the means. Therefore, there were no significant between-gender differences in the distribution of results or the reference intervals, although the results tended to be somewhat lower in women compared to men, as previously noted [1].

Of the 6892 PREVEND participants that completed the second round of screening, 4336 subjects were included in this study. Subjects were excluded if they were missing NMR data, if they were missing information regarding T2DM diagnosis at baseline or follow-up, or if they were diagnosed with T2DM at baseline. Baseline clinical and laboratory characteristics for the entire cohort as well as according to tertiles of betaine are shown in Table 3. Subjects with higher betaine levels were older, more likely to be men and less likely to be Caucasian. Subjects with higher betaine levels also had lower BMI, were more likely to be non-smokers and less likely to have high diastolic blood pressure. Subjects in the highest tertile of betaine had lower, total cholesterol, TG and C-reactive protein (CRP). They were more likely to be on lipid-lowering medications. 

After a median (interquartile range) follow up period of 7.3 (IQR, 5.9–7.7) years, 224 new T2DM cases were ascertained. Cox proportional hazards regression models revealed that the highest tertile of betaine was associated with a lower incidence of T2DM. Hazard ratio (HR) for the crude model was 0.61 (95% CI: 0.44–0.85, *p* = 0.004) (Table 4). The association remained largely unchanged after adjustment for multiple clinical covariates and T2DM risk factors, including age, sex, eGFR, BMI, smoking status, ethnicity, FPG, total cholesterol, HDL-C, TG, CRP, and use of lipid-lowering drugs. In the fully adjusted model, the HR for the upper vs. the lower tertile was 0.50 (95% CI: 0.32–0.80, *p* = 0.004). When the association of betaine with T2DM development was assessed as a continuous variable, the same trend was observed albeit the association of betaine with incident T2DM did not reach statistical significance in the fully adjusted model. 

We observed that the association of betaine with T2DM was modified by sex (P_interaction_ = 0.01), but not by age (P_interaction_ = 0.93) or eGFR (P_interaction_ = 0.38). Betaine as continuous variable was inversely associated with incident T2DM in men (HR: 0.44, 95% CI: 0.24–0.80, *p* = 0.007), but not in women (HR: 0.97, 95% CI: 0.55–1.71, *p* = 0.93), after adjustment for age, eGFR, BMI, smoking status, ethnicity, FPG, total cholesterol, HDL-C, TG, CRP, and use of lipid-lowering drugs (Table 5). 

To ensure the results would be similar if the analyses were conducted in subjects at a lower risk of CVD, a sensitivity analysis was performed excluding subjects with previous CVD, use of lipid-lowering drugs, microalbuminuria and eGFR <60 mL/min/1.73 m^2^ at baseline (*n* = 2810 subjects with 112 cases of T2DM ascertained) (Table 6). Cox proportional hazards regression models revealed that the highest tertile of betaine was associated with a lower incidence of T2DM. HR for the crude model was 0.58 (95% CI: 0.36–0.93, *p* = 0.03). The association remained significant after adjustment for multiple clinical covariates and T2DM risk factors, including age, sex, eGFR, BMI, and smoking status. Only the fully-adjusted model, with additional cumulative adjustment for ethnicity, FPG, total cholesterol, HDL-C, TG, and CRP, lost statistical significance (0.57 95% CI: 0.32–1.04, *p* = 0.07).

## 4. Discussion

Several chromatographic, chemical, and mass spectrometry (MS)-based techniques for quantification of betaine have been reported [10,32,33,39,40]. With the advent of the Vantera clinical analyzer, an automated high-throughput NMR spectrometer, NMR-based assays are now available in the diagnostic laboratory for clinical use [29,30,41,42,43,44]. Hence, an NMR-based assay that quantifies plasma/serum TMAO and betaine was developed for the diagnostic laboratory [4,21,29]. While the NMR signals for TMAO and betaine align in NMR spectra acquired at physiological pH, this assay uses a buffer with a lower pH that separates the two metabolites, allowing for their simultaneous quantification [29]. Analytical validation of the NMR-based betaine assay revealed that it has performance characteristics that are robust enough for use as a diagnostic or prognostic test in the clinical laboratory.

The major finding of this study, however, is that lower circulating levels of betaine are associated with future development of T2DM in a large population-based cohort. Lower betaine levels were associated with increased risk of developing T2DM even after adjusting for clinical characteristics as well as typical risk factors for T2DM such as BMI, lipids, and FPG. Previously, lower betaine levels were reported to be associated with incident T2DM development in subjects across a spectrum of risks for CVD, such as in Norwegian subjects with suspected angina pectoris, as well as in primary prevention individuals enrolled in the PREDIMED dietary intervention trial [7,28]. In addition, betaine was associated with future diabetes in the DPP, a program designed to study progression to, and possibly prevention of, T2DM in subjects at high risk of T2DM [6]. Moreover, the prospective association of low betaine levels with future T2DM in DPP remained significant after adjusting for risk factors such as age, sex, BMI, hypertension, ethnicity and FPG [6]. Furthermore, the DPP lifestyle intervention led to an increase in betaine, raising the hypothesis that betaine may have utility in monitoring the effects of interventions designed to prevent progression to T2DM [6]. 

Betaine is required for remethylation of homocysteine to methionine, a precursor of the universal methyl donor SAM [1,2]. By decreasing SAM availability, betaine deficiency may decrease phosphatidylcholine synthesis, promote hepatic steatosis, and modify VLDL synthesis and secretion [3]. Betaine was shown to be inversely associated with TG and PLTP activity, further supporting the notion that low betaine levels may alter liver fat accumulation and lipid/lipoprotein metabolism [4]. Furthermore, lower plasma betaine concentrations have been reported in subjects with metabolic syndrome, T2DM, NAFLD, and/or NASH, supporting the association of low levels of betaine with diseases that are associated with liver fat accumulation [3,5,6,7,8]. Moreover, betaine levels were associated with disease severity in subjects with NAFLD, betaine levels being significantly lower in subjects with NASH compared to subjects with hepatic steatosis [3]. These observations support the findings of this study, that lower betaine levels may be a good predictor of future T2DM. 

Given that betaine has been described as being a precursor to TMAO, it seems counterintuitive that lower betaine levels would coincide with higher TMAO levels and vice versa. However, we did not find a statistically significant correlation between TMAO and betaine in the normal healthy population we used for our reference interval study or in the PREVEND study. This may be attributed to the fact that there are other precursors for TMAO such as carnitine [11], choline, phosphatidylcholine [10], and trimethyllysine [45] that can give rise to TMAO via trimethylamine. In addition, choline is oxidized to betaine [27] in humans, and betaine can be shunted for multiple metabolic functions [1,2] Therefore, there may not be a clear relationship between TMAO and betaine, especially in normal, healthy populations such as the ones analyzed in this study.

Subjects with cardiometabolic diseases such as T2DM and non-alcoholic fatty liver disease (NAFLD) exhibit altered gut microbiome, and have higher levels of TMAO and lower levels of betaine compared to healthy subjects [3,5,6,7,8,20,25,46,47]. Betaine, the gut-microbiome generated metabolite TMAO, and dietary metabolites such as choline and L-carnitine, may help determine if a subject’s gut microbiome is altered (gut dysbiosis), and identify individuals who may benefit from intensive dietary or lifestyle intervention in order to reduce the progression to more severe metabolic disease such as T2DM. Further studies are needed to confirm this to be the case. 

There are several strengths and limitations in the current study. A strength of this study is that it includes a large number of participants in the general population, whereas previously published studies reported associations of betaine with incident T2DM in subjects who may have been at risk of CVD or T2DM. In addition, the PREVEND study has a large number of participants with a large age range, which could be considered a strength. Notably, the PREVEND study was designed to study the impact of albuminuria on renal and CVD outcomes. Hence, the PREVEND subjects were preferentially recruited based on an initially elevated urinary albumin concentration. However, a sensitivity analysis excluding subjects with microalbuminuria and compromised renal function showed an essentially similar association of betaine with future risk of T2DM. Furthermore, the majority of the PREVEND participants were of Caucasian descent, and study results may not be applicable to subjects of non-white ethnicities.

## 5. Conclusions

The newly developed NMR-based betaine assay exhibits excellent performance characteristics. We found that lower circulating betaine was associated with increased risk of developing T2DM in a large population-based cohort even after adjusting for clinical covariates and T2DM risk factors. Potential clinical applications may include assessing the risk of future T2DM.

## Figures and Tables

**Figure 1 jcm-08-01813-f001:**
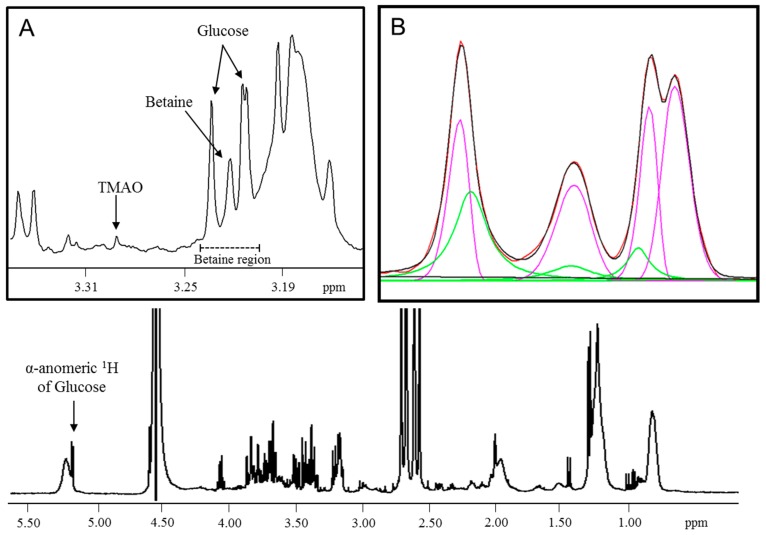
1D 1H CPMG NMR spectrum of serum used for modeling the betaine peak. (**A**) Expansion of the spectrum where the betaine region is extracted for lineshape deconvolution. (**B**) The experimental (red) spectrum overlaid with a mathematical fit (black) from a composite of lognormal (pink) and Lorentzian (green) lineshapes after subtraction of the baseline in the betaine region. ppm = parts per million; CPMG, Carr–Purcell–Meiboom–Gill; NMR, nuclear magnetic resonance.

**Figure 2 jcm-08-01813-f002:**
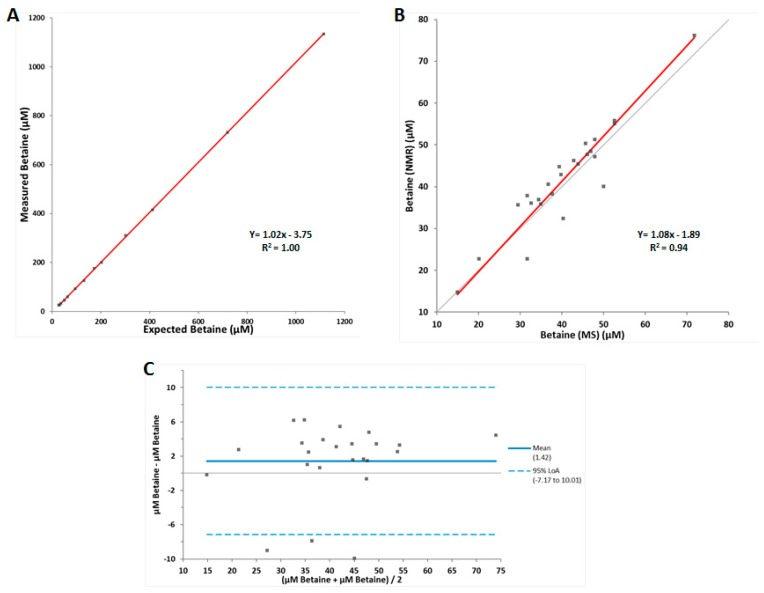
Linearity and method comparison data for the betaine assay. **(A**) Results of linearity study (*n* = 13), (**B**) Deming regression comparison between LC-MS/MS and NMR measured betaine in serum samples (*n* = 24), (**C**) Bland–Altman plot (*n* = 24). The limits of agreement (LoA) are depicted as dotted blue lines, and the 0% bias is a solid black line.

**Table 1 jcm-08-01813-t001:** Within-laboratory (inter-assay) and within-run (intra-assay) imprecision.

Imprecision	Betaine (µM)		
	Low	Medium	High
Within-lab ^a^			
Mean	45.2	97.2	205.9
SD	2.5	4.0	5.2
CV (%)	5.5	4.1	2.5
Within-run ^b^			
Mean	44.1	95.6	205.6
SD	1.9	3.1	3.1
CV (%)	4.3	3.2	1.5

Abbreviations: CV, coefficients of variation; SD, standard deviation; ^a^ Based on CLSI EP5-A2 tested using three controls, two runs per day in duplicate for 20 days (total *n* = 80); ^b^ Based on one run of 20 tests.

**Table 2 jcm-08-01813-t002:** Distribution and population means for betaine (µM) in generally healthy adults.

Percentile	Normal Healthy Adult Volunteers (*n* = 501)	Normal Healthy Adult Volunteers Female (*n* = 292)	Normal Healthy Adult Volunteers Male (*n* = 209)	PREVEND Study Participants (*n* = 5621)
0th	<13.2 ^a^	<13.2 ^a^	24.6	13.3
2.5th	23.8	22.0	28.5	21.0
25.0th	34.1	32.4	37.3	30.8
50.0th	41.0	39.1	44.6	36.8
75.0th	49.5	46.9	52.4	43.8
97.5th	74.7	74.4	75.3	63.0
100th	104.1	101.6	104.1	190.7
Mean (SD)	42.9 (12.6)	40.7 (12.4)	46.0 (12.2)	38.1 (11.2)

Abbreviations: SD, standard deviation. ^a^ Lower bound of the reportable range is limited by the LOQ of this assay.

**Table 3 jcm-08-01813-t003:** Baseline characteristics of the 4336 subjects of the Prevention of Renal and Vascular End-Stage Disease (PREVEND) study of the overall population and according to tertiles of betaine.

	Overall	Tertiles of betaine			*p*-Value
		T1	T2	T3	
Participants, *n*	4336	1445	1446	1445	
Betaine, µM	36.9 (31.0–44.0)	28.7 (25.3–31.0)	36.9 (35.1–38.8)	47.3 (44.0–52.6)	
Participants, *n*	4236	1398	1418	1420	
TMAO, µM	4.6 ± 5.9	4.6 ± 5.1	4.7 ± 6.3	4.6 ± 6.4	
**General characteristics**					
Age, years	52.6 ± 11.5	51.3 ± 10.9	52.8 ± 11.4	53.7 ± 12.1	<0.001
Male sex, *n* (%)	2159 (49.8)	456 (31.6)	750 (51.9)	953 (66.0)	<0.001
Ethnicity, Caucasian, *n* (%)	4164 (96.0)	1403 (97.1)	1400 (96.8)	1361 (94.2)	0.001
BMI, kg/m^2^	26.4 ± 4.2	26.7 ± 4.3	26.6 ± 4.2	26.0 ± 4.1	<0.001
Smoking status, *n* (%)					0.008
Never	1286 (29.7)	420 (29.1)	424 (29.3)	442 (30.6)	
Former	1828 (42.2)	574 (39.7)	639 (44.2)	615 (42.6)	
Current	1165 (26.9)	438 (30.3)	358 (24.8)	369 (25.5)	
Alcohol consumption, never, *n* (%)	1004 (23.2)	329 (22.8)	351 (24.3)	324 (22.4)	0.26
eGFR, mL/min/1.73m^2^	93.5 ± 16.3	94.5 ± 15.9	93.1 ± 16.7	93.0 ± 16.3	0.02
Hypertension, *n* (%)	1280 (29.5)	406 (28.1)	443 (30.6)	431 (29.8)	0.33
Hypercholesterolemia, *n* (%)	1235 (28.5)	468 (32.4)	390 (27.0)	377 (26.1)	<0.001
Parental history of CKD, *n* (%)	20 (0.5)	8 (0.6)	7 (0.5)	5 (0.3)	0.70
**Circulation**					
SBP, mmHg	124.8 ± 18.1	123.9 ± 17.8	125.4 ± 18.2	125.1 ± 18.3	0.06
DBP, mmHg	73.0 ± 9.0	72.5 ± 9.1	73.4 ± 8.8	72.3 ± 9.0	0.02
**Laboratory parameters**					
Total cholesterol, mmol/L	5.4 ± 1.0	5.6 ± 1.1	5.4 ± 1.0	5.2 ± 1.0	<0.001
HDL cholesterol, mmol/L	1.3 ± 0.3	1.2 ± 0.3	1.3 ± 0.3	1.2 ± 0.3	<0.001
Triglycerides, mmol/L	1.1 (0.8–1.6)	1.2 (0.8–1.7)	1.1 (0.8–1.6)	1.0 (0.8–1.5)	0.005
Fasting glucose, mmol/L	4.7 (4.4–5.2)	4.7 (4.4–5.2)	4.7 (4.4–5.2)	4.7 (4.4–5.2)	0.34
C-reactive protein, mg/L	1.2 (0.6–2.8)	1.4 (0.6–3.0)	1.2 (0.5–2.6)	1.1 (0.5–2.8)	0.005
**Medication**					
Antihypertensive drugs, *n* (%)	716 (16.5)	221 (15.3)	239 (16.5)	256 (17.7)	0.12
Lipid lowering drug use, *n* (%)	304 (7.0)	62 (4.3)	94 (6.5)	148 (10.2)	<0.001

Data are represented as mean ± SD, median (interquartile range) or *n* (%). Differences were tested by ANOVA or Kruskal Wallis for continuous variables and with χ^2^- test for categorical variables. The eGFR is based on the creatinine–cystatin C equation. Abbreviations: BMI, body mass index; eGFR, estimated glomerular filtration rate; CKD, chronic kidney disease; SBP, systolic blood pressure; DBP, diastolic blood pressure; HDL, high-density lipoproteins.

**Table 4 jcm-08-01813-t004:** Association of betaine as a continuous variable and according to tertiles with the development of T2DM.

	Betaine as Continuous Variable (^2^log)		Tertiles of Betaine				
			T1	T2		T3	
	HR (95% CI)	*p*		HR (95% CI)	*p*	HR (95% CI)	*p*
Diabetes, no. of events (%)	224 (5.2%)		93 (6.4%)	74 (5.1%)		57 (3.9%)	
Crude	0.79 (0.59–1.05)	0.10	1.00 (ref)	0.78 (0.57–1.05)	0.10	0.61 (0.44–0.85)	0.004
Model 1	0.60 (0.46–0.79)	<0.001	1.00 (ref)	0.64 (0.47–0.88)	0.005	0.45 (0.32–0.64)	<0.001
Model 2	0.59 (0.45–0.78)	<0.001	1.00 (ref)	0.61 (0.45–0.84)	0.002	0.42 (0.29–0.59)	<0.001
Model 3	0.63 (0.47–0.85)	0.002	1.00 (ref)	0.68 (0.50–0.94)	0.02	0.47 (0.33–0.66)	<0.001
Model 4	0.69 (0.46–1.02)	0.06	1.00 (ref)	0.65 (0.43–0.96)	0.03	0.50 (0.32–0.80)	0.004

Abbreviations: eGFR, estimated glomerular filtration rate; T2DM, type 2 diabetes mellitus. Association between betaine and development of diabetes in 4336 (224 cases) subjects of the Prevention of Renal and Vascular End-Stage Disease (PREVEND) study as a continuous variable (^2^log-transformed) and according to tertiles (T1-T3). Hazard ratios and 95% confidence intervals were derived from Cox proportional hazards regression models. The eGFR is based on creatinine–cystatin C equation; Model 1: Adjustment for age and sex; Model 2: Model 1+ adjustment for eGFR; Model 3: Model 2 + adjustment for body mass index and smoking; Model 4: Model 3 + adjustment for ethnicity, fasting glucose, total cholesterol, high-density lipoprotein cholesterol, triglycerides, C-reactive protein, and use of lipid-lowering drugs.

**Table 5 jcm-08-01813-t005:** Sex-stratified analyses of the association of betaine as a continuous variable and according to tertiles with the development of T2DM.

	Betaine as Continuous Variable (^2^log)		Tertiles of Betaine				
			T1	T2		T3	
	HR (95% CI)	*p*		HR (95% CI)	*p*	HR (95% CI)	*p*
Men							
Diabetes, no. of events (%)	139 (6.4%)		54 (2.5%)	46 (2.1%)		39 (1.8%)	
Crude	0.41 (0.26–0.64)	<0.001	1.00 (ref)	0.49 (0.33–0.73)	<0.001	0.34 (0.22–0.51)	<0.001
Model 1	0.40 (0.26–0.62)	<0.001	1.00 (ref)	0.49 (0.33–0.72)	<0.001	0.32 (0.21–0.49)	<0.001
Model 2	0.41 (0.26–0.65)	<0.001	1.00 (ref)	0.46 (0.31–0.69)	<0.001	0.32 (0.21–0.48)	<0.001
Model 3	0.56 (0.35–0.90)	0.02	1.00 (ref)	0.54 (0.36–0.82)	0.003	0.43 (0.28–0.65)	<0.001
Model 4	0.44 (0.24–0.80)	0.007	1.00 (ref)	0.43 (0.25–0.74)	0.003	0.36 (0.20–0.65)	0.001
Women							
Diabetes, no. of events (%)	85 (3.9%)		39 (1.8%)	28 (1.3%)		18 (0.8%)	
Crude	0.96 (0.59–1.53)	0.85	1.00 (ref)	1.01 (0.62–1.63)	0.98	0.95 (0.54–1.66)	0.86
Model 1	0.81 (0.52–1.27)	0.35	1.00 (ref)	0.89 (0.55–1.45)	0.65	0.83 (0.47–1.45)	0.51
Model 2	0.74 (0.48–1.15)	0.18	1.00 (ref)	0.86 (0.53–1.41)	0.56	0.67 (0.37–1.22)	0.19
Model 3	0.73 (0.47–1.15)	0.18	1.00 (ref)	0.96 (0.58–1.56)	0.85	0.62 (0.33–1.15)	0.13
Model 4	0.97 (0.55–1.71)	0.93	1.00 (ref)	1.02 (0.56–1.89)	0.94	0.74 (0.35–1.60)	0.45

Abbreviations: eGFR, estimated glomerular filtration rate; T2DM, type 2 diabetes mellitus. Association between betaine and development of diabetes in 2159 men (139 cases) and 2177 women (85 cases) of the Prevention of Renal and Vascular End-Stage Disease (PREVEND) study as a continuous variable (^2^log-transformed) and according to tertiles (T1-T3). Hazard ratios and 95% confidence intervals were derived from Cox proportional hazards regression models. The eGFR is based on creatinine–cystatin C equation; Model 1: Adjustment for age; Model 2: Model 1+ adjustment for eGFR; Model 3: Model 2 + adjustment for body mass index and smoking; Model 4: Model 3 + adjustment for ethnicity, fasting glucose, total cholesterol, high-density lipoprotein cholesterol, triglycerides, C-reactive protein, and use of lipid-lowering drugs.

**Table 6 jcm-08-01813-t006:** Sensitivity analysis of the association of betaine as a continuous variable and according to tertiles with development of T2DM, excluding subjects with previous CVD history, microalbuminuria, eGFR < 60 mL/min/1.73 m^2^, and use of lipid-lowering drugs.

	Betaine as Continuous Variable (^2^log)		Tertiles of Betaine				
			T1	T2		T3	
	HR (95% CI)	*p*		HR (95% CI)	*p*	HR (95% CI)	*p*
Diabetes, no. of events (%)	112 (4.0%)		48 (4.9%)	39 (4.2%)		25 (2.8%)	
Crude	0.75 (0.51–1.10)	0.14	1.00 (ref)	0.84 (0.55–1.28)	0.41	0.58 (0.36–0.93)	0.03
Model 1	0.57 (0.41–0.81)	0.002	1.00 (ref)	0.69 (0.45–1.05)	0.09	0.41 (0.25–0.67)	<0.001
Model 2	0.59 (0.40–0.80)	0.001	1.00 (ref)	0.69 (0.45–1.05)	0.08	0.41 (0.25–0.67)	<0.001
Model 3	0.62 (0.42–0.93)	0.02	1.00 (ref)	0.75 (0.49–1.15)	0.19	0.50 (0.31–0.82)	0.006
Model 4	0.69 (0.43–1.10)	0.12	1.00 (ref)	0.89 (0.55–1.46)	0.65	0.57 (0.32–1.04)	0.07

Abbreviations: CVD, cardiovascular disease; eGFR, estimated glomerular filtration rate; T2DM, type 2 diabetes mellitus. Association between betaine and development of diabetes in 2810 (112 cases) subjects of the Prevention of Renal and Vascular End-Stage Disease (PREVEND) study as a continuous variable (^2^log-transformed) and according to tertiles (T1-T3). Hazard ratios and 95% confidence intervals were derived from Cox proportional hazards regression models. The eGFR is based on creatinine–cystatin C equation; Model 1: Adjustment for age and sex; Model 2: Model 1+ adjustment for Egfr; Model 3: Model 2 + adjustment for body mass index and smoking; Model 4: Model 3 + adjustment for ethnicity, fasting glucose, total cholesterol, high-density lipoprotein cholesterol, triglycerides, and C-reactive protein.

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
