# Peer review of "High Betaine, a Trimethylamine N-Oxide Related Metabolite, Is Prospectively Associated with Low Future Risk of Type 2 Diabetes Mellitus in the PREVEND Study"

_jcm, 2019, doi:10.3390/jcm8111813_

Round 1

Reviewer 1 Report

Manuscript jcm-588318 entitled “High Betaine, a Trimethylamine N-oxide Related Metabolite, is Prospectively Associated with Low Future Risk of Type 2 Diabetes Mellitus in the PREVEND Study” reports a newly developed NMR-based betaine assay that exhibits performance characteristics consistent with usage in the clinical laboratory for assessing risk of future T2DM continuing previous work by the team. The manuscript is clearly written, findings are robust and valuable to the field.

Minor comments:

Please clarify in the introduction the relation between betaine and TMAO. Since their relation with cardiometabolic diseases is inverse ( decreased betaine leads to increased T2DM risk and Increased TMAO leads to increased CV risk) and, being betaine a precursor of TMAO, it is not clear if lower betaine levels decrease TMAO or not and what is the relation between the 2 biomarkers in the population assessed (n=501). The manuscript would benefit from clarification of the relation between betaine and TMAO and from presenting the results of TMAO quantification in the population where accurate TMAO quantification was possible. Results of TMAO quantification should be presented and discussed, since the methodology used allows it.

Author Response

Comments and Suggestions for Authors

Manuscript jcm-588318 entitled “High Betaine, a Trimethylamine N-oxide Related Metabolite, is Prospectively Associated with Low Future Risk of Type 2 Diabetes Mellitus in the PREVEND Study” reports a newly developed NMR-based betaine assay that exhibits performance characteristics consistent with usage in the clinical laboratory for assessing risk of future T2DM continuing previous work by the team. The manuscript is clearly written, findings are robust and valuable to the field.

Response: The authors would like to thank the reviewer for her/his time and for stating that the findings of our current work are valuable to the field.

Minor comments:

Please clarify in the introduction the relation between betaine and TMAO. Since their relation with cardiometabolic diseases is inverse ( decreased betaine leads to increased T2DM risk and Increased TMAO leads to increased CV risk) and, being betaine a precursor of TMAO, it is not clear if lower betaine levels decrease TMAO or not and what is the relation between the 2 biomarkers in the population assessed (n=501). The manuscript would benefit from clarification of the relation between betaine and TMAO and from presenting the results of TMAO quantification in the population where accurate TMAO quantification was possible. Results of TMAO quantification should be presented and discussed, since the methodology used allows it. 

Response: We agree with the reviewer that this is an important clarification to make. We modified appropriate sections of the manuscript. Because the focus of the paper is on betaine and not its relationships to TMAO, we modified the first sentence of the abstract to read: “Gut microbiota related metabolites trimethylamine-N-oxide (TMAO), choline and betaine have been shown to be associated with cardiovascular disease (CVD) risk.” and removed the reference to betaine being a precursor of TMAO.  Hopefully this will alleviate any confusion to readers of the publication.  We also presented the results of TMAO quantification in Table 3. We did however, add a paragraph in the discussion section to clarify any potential relationship between betaine and TMAO.  This paragraph reads:  “Given that betaine has been described as being a precursor to TMAO, it seems counterintuitive that lower betaine levels would coincide with higher TMAO levels and vice versa. However, we did not find a statistically significant correlation between TMAO and betaine the normal healthy population we used for our reference interval study or in the PREVEND study. This may be attributed to the fact that there are other precursors for TMAO such as carnitine [11], choline, phosphatidylcholine [10] and trimethyllysine [45] that can give rise to TMAO via trimethylamine. In addition, choline is oxidized to betaine [27] in humans and betaine can be shunted for multiple metabolic functions [1,2] Therefore, there may not be a clear relationship between TMAO and betaine, especially in normal, healthy populations such as the ones analyzed in this study.”  In addition, we looked for correlations between betaine and TMAO in both populations used in our study and found no statistically significant correlations in either the normal healthy population study that was used for the reference interval determination (r = 0.065, p = 0.149) or the PREVEND study (r = -0.01, p = 0.508).  We added sentences to this effect in the methods and discussions sections.

Reviewer 2 Report

The authors present an interesting, concise and well-written study to determine the usefulness of a newly NMR-based betaine assay and at the same time, to apply it for discovering potential associations between betaine levels and T2DM.

I have only some comments:

In Assay Performance Testing section:

“Within-run (n=20 replicates) and within-laboratory (n=80 replicates) imprecision were determined using serum pools with low, intermediate and high betaine values.” Comment: How the athours stablish what low, intermediate and high betaine values are?

“The LC-MS/MS method employed an AB SCIEX 5000 triple quadrupole mass spectrometer using d9-(trimethyl)-labelled internal standards.” Comment: Since the authors gave details about the mass spectrometers, maybe the also should included the HPLC and the column.

In Results section:

“(Pinteracton = 0.01), but not by age (Pinteraction=0.93) or eGFR (Pinteraction=0.38)”. Comment: Please, mantain the same format.

In Discussion:

“Subjects with cardiometabolic diseases such as T2DM and non-alcoholic fatty liver disease (NAFLD) exhibit altered gut microbiome, and have higher levels of TMAO and lower levels of betaine compared to healthy subjects [3,5-8,20,25,45,46]. Betaine, the gut-microbiome generated metabolite TMAO, and dietary metabolites such as choline and L-carnitine, may help determine if a subject’s gut microbiome is altered (gut dysbiosis), and identify individuals who may benefit from intensive dietary or lifestyle intervention in order to reduce the progression to more severe metabolic disease such as T2DM. Further studies are needed to confirm this to be the case.” Comment: this is the only explation provided from authors about why lower betaine levels are associated with T2DM. Even if I understand that it is not the main goal of the research, I miss some deeper explanation about that relationship, because the authors gave a lot importance to it.

4. Conclusions section can be improved. Since the title of the manuscript is the association between betaine and T2DM, and the authors declare that this is the main result extracted from the study, the conclusions should include it.

Author Response

Comments and Suggestions for Authors

The authors present an interesting, concise and well-written study to determine the usefulness of a newly NMR-based betaine assay and at the same time, to apply it for discovering potential associations between betaine levels and T2DM.

Response: We appreciate the reviewer for the thorough review of and critical comments to improve the manuscript.

I have only some comments:

In Assay Performance Testing section:

“Within-run (n=20 replicates) and within-laboratory (n=80 replicates) imprecision were determined using serum pools with low, intermediate and high betaine values.” Comment: How the athours stablish what low, intermediate and high betaine values are?

Response:  We added the sentence “These values were targeted around the 50th and 100th percentile of the reference interval, as well a high value that had been observed in clinical samples.”

“The LC-MS/MS method employed an AB SCIEX 5000 triple quadrupole mass spectrometer using d9-(trimethyl)-labelled internal standards.” Comment: Since the authors gave details about the mass spectrometers, maybe the also should included the HPLC and the column.

Response: We modified this sentence to read the following: “Samples on the LC-MS/MS method were injected at a flowrate of 0.8 ml/min to a 4.6 x 250 mm, 5 µm Luna silica column interfaced with an AB SCIEX 5000 triple quadrupole mass spectrometer using d9-(trimethyl)-labelled internal standards [32].”, and appended the reference for a detailed description of the analytical setup.

In Results section:

“(Pinteracton = 0.01), but not by age (Pinteraction=0.93) or eGFR (Pinteraction=0.38)”. Comment: Please, mantain the same format.

Response: We thank the reviewer for this comment. We have now used Pinteraction throughout the manuscript.

In Discussion:

“Subjects with cardiometabolic diseases such as T2DM and non-alcoholic fatty liver disease (NAFLD) exhibit altered gut microbiome, and have higher levels of TMAO and lower levels of betaine compared to healthy subjects [3,5-8,20,25,45,46]. Betaine, the gut-microbiome generated metabolite TMAO, and dietary metabolites such as choline and L-carnitine, may help determine if a subject’s gut microbiome is altered (gut dysbiosis), and identify individuals who may benefit from intensive dietary or lifestyle intervention in order to reduce the progression to more severe metabolic disease such as T2DM. Further studies are needed to confirm this to be the case.” Comment: this is the only explation provided from authors about why lower betaine levels are associated with T2DM. Even if I understand that it is not the main goal of the research, I miss some deeper explanation about that relationship, because the authors gave a lot importance to it.

Response:  We concur that the discussion section would benefit by an explanation of why lower betaine levels may be associated with T2DM and added the following paragraph:  “Betaine is required for remethylation of homocysteine to methionine, a precursor of the universal methyl donor SAM [1,2]. By decreasing SAM availability, betaine deficiency may decrease phosphatidylcholine synthesis, promote hepatic steatosis and modify VLDL synthesis and secretion [3]. Betaine was shown to be inversely associated with TG and PLTP activity, further supporting the notion that low betaine levels may alter liver fat accumulation and lipid/lipoprotein metabolism [4]. Furthermore, lower plasma betaine concentrations have been reported in subjects with metabolic syndrome, T2DM, NAFLD and/or NASH, supporting the association of low levels of betaine with diseases that are associated with liver fat accumulation [3,5-8]. Moreover, betaine levels were associated with disease severity in subjects with NAFLD; betaine levels being significantly lower in subjects with NASH compared to subjects with hepatic steatosis [3]. These observations support the findings of this study, that lower betaine levels may be a good predictor of future T2DM.”

Conclusions section can be improved. Since the title of the manuscript is the association between betaine and T2DM, and the authors declare that this is the main result extracted from the study, the conclusions should include it.

Response: We appreciate the suggestion of the reviewer. We added “We found that lower circulating betaine were associated with increased risk of developing T2DM in a large population-based cohort even after adjusting clinical covariates and T2DM risk factors.” in the conclusions section.